# *Bacillus thuringiensis* Cry4Ba Insecticidal ToxinExploits Leu^615^ in Its C-Terminal Domain to Interact with a Target Receptor—*Aedes aegypti* Membrane-Bound Alkaline Phosphatase

**DOI:** 10.3390/toxins13080553

**Published:** 2021-08-09

**Authors:** Anon Thammasittirong, Sutticha Na-Ranong Thammasittirong, Chompounoot Imtong, Sathapat Charoenjotivadhanakul, Somsri Sakdee, Hui-Chun Li, Siriporn Okonogi, Chanan Angsuthanasombat

**Affiliations:** 1Microbial Biotechnology Unit, Department of Microbiology, Faculty of Liberal Arts and Science, Kasetsart University, Nakhon Pathom 73140, Thailand; sutticha.n@ku.ac.th; 2Faculty of Science and Technology, Prince of Songkla University, Pattani 94000, Thailand; chompou_ron@hotmail.com; 3Bacterial Toxin Research Innovation Cluster (BRIC), Institute of Molecular Biosciences, Salaya Campus, Mahidol University, Nakorn Pathom 73170, Thailand; sathapat.dua@student.mahidol.ac.th (S.C.); sakdees@gmail.com (S.S.); 4Department of Biochemistry, School of Medicine, Tzu Chi University, Hualien 97004, Taiwan; huichun@gms.tcu.edu.tw; 5Research Center of Pharmaceutical Nanotechnology, Department of Pharmaceutical Sciences, Faculty of Pharmacy, Chiang Mai University, Chiang Mai 50200, Thailand; okng2000@gmail.com; 6Laboratory of Synthetic Biophysics and Chemical Biology, Biophysics Institute for Research and Development (BIRD), Chiang Mai 50130, Thailand

**Keywords:** Cry4Ba mosquito-active toxin, enzyme-linked immunosorbent assay, GPI-anchored alkaline phosphatase, homology-based modeling, molecular docking, receptor binding

## Abstract

In addition to the receptor-binding domain (DII), the C-terminal domain (DIII) of three-domain Cry insecticidal δ-endotoxins from *Bacillus thuringiensis* has been implicated in target insect specificity, yet its precise mechanistic role remains unclear. Here, the 21 kDa high-purity isolated DIII fragment derived from the Cry4Ba mosquito-specific toxin was achieved via optimized preparative FPLC, allowing direct rendering analyses for binding characteristics toward its target receptor—*Aedes aegypti* membrane-bound alkaline phosphatase (Aa-mALP). Binding analysis via dotblotting revealed that the Cry4Ba-DIII truncate was capable of specific binding to nitrocellulose-bound Aa-mALP, with a binding signal comparable to its 65 kDa Cry4Ba-R203Q full-length toxin. Further determination of binding affinity via sandwich ELISA revealed that Cry4Ba-DIII exhibited a rather weak binding to Aa-mALP with a dissociation constant (*K*_d_) of ≈1.1 × 10^−7^ M as compared with the full-length toxin. Intermolecular docking between the Cry4Ba-R203Q active toxin and Aa-mALP suggested that four Cry4Ba-DIII residues, i.e., Glu^522^, Asn^552^, Asn^576^, and Leu^615^, are potentially involved in such toxin–receptor interactions. Ala substitutions of each residue (E522A, N552A, N576A and L615A) revealed that only the L615A mutant displayed a drastic decrease in biotoxicity against *A. aegypti* larvae. Additional binding analysis revealed that the L615A-impaired toxin also exhibited a reduction in binding capability to the surface-immobilized Aa-mALP receptor, while two bio-inactive DII-mutant toxins, Y332A and F364A, which almost entirely lost their biotoxicity, apparently retained a higher degree of binding activity. Altogether, our data disclose a functional importance of the C-terminal domain of Cry4Ba for serving as a potential receptor-binding moiety in which DIII-Leu^615^ could conceivably be exploited for the binding to Aa-mALP, highlighting its contribution to toxin interactions with such a target receptor in mediating larval toxicity.

## 1. Introduction

Currently, several strains of *Bacillus thuringiensis* (*Bt*), a Gram-positive, spore-forming aerobic bacterium producing various insecticidal crystal proteins (primarily known as Cry δ-endotoxins), have been wildly used as safe bio-insecticides for the control of agricultural pests and human disease vectors [1,2,3]. These *Bt*-crystal proteins are specifically toxic to particular types of target insect larvae but harmless to both humans and other non-target organisms [2,3]. For example, the Cry4Ba δ-endotoxin produced from *Bt* subsp. *israelensis* (*Bti*) is highly active against the larvae of *Aedes* and *Anopheles* spp., which are mosquito vectors of important tropical infectious diseases such as dengue hemorrhagic fever, chikungunya, yellow fevers, and malaria [4,5].

Upon ingestion by larvae of a susceptible insect species, individual Cry toxins that are principally produced as protoxin inclusions (e.g., the ≈130 kDa Cry4Ba mosquito-specific protoxin) are dissolved in alkaline midgut fluid and then proteolytically processed by gut proteases to yield active toxins of ≈65 kDa [1,6]. In general, the activated Cry toxins display a typical wedge-shaped arrangement of three distinctive domains: an N-terminal domain of eight α-helices (DI), a three-β-sheet domain (DII), and a C-terminal domain of β-sheet sandwich (DIII) [7] (see Figure 1). Subsequently, the activated toxins bind specifically to target receptors lining the brush-border membrane of larval midgut epithelial cells [8], followed by toxin insertion into the target cell membrane to form an ion-leakage pore, which eventually results in midgut cell lysis. Disruption of the gut epithelium would lead to starvation and eventually to death of the intoxicated larvae [9]. Nonetheless, structural details of toxic mechanism underlying specific effects of individual Cry toxins still remain to be further explored. Recently, we have provided detailed structural insights into Cry4Ba-induced lytic pore formation by demonstrating that a membrane-bound state of toxin monomers is a critical prerequisite for the assembly of a potential functional pre-pore trimer [10].

To date, both DI (the N-terminal α-helical bundle) and DII (a β-sheet prism structure with several exposed loops) of numerous Cry toxins have been evidently demonstrated to play a pivotal role in membrane-inserted pore formation and target receptor recognition, respectively [7,9,11,12]. Of particular significant findings in DI for the pore-lining α4-loop-α5 hairpin, we have shown that the polarity of the Cry4Ba α4-α5 loop residue‒Asn^166^ was critically involved in ion permeation through the toxin-induced pore, which is likely to facilitate the toxin-pore opening [13]. We have also disclosed the functional importance of the intrinsic stability toward the Pro-rich cluster (Pro^193^Pro^194^_Pro^196^), which is present only in the long loop linking two pore-lining helices (α4 and α5) of Cry4Aa, another *Bti* toxin closely related to Cry4Ba [14]. Very recently, we have signified a critical involvement in Cry4Aa biotoxicity of His^180^ present in the pore-lumen-facing α4, revealing that an adequate size of this side-chain is likely crucial for supporting the conserved hydrophobic core found within the Cry4Aa-DI, thus conceivably providing suitable surroundings for the α4-α5 hairpin prior to membrane-inserted pore formation [15]. For a functional role in receptor recognition of DII, although most other studies are restricted to only three β-hairpin loops, i.e., β2-β3, β6-β7, and β10-β11loops, we have shown that two other Cry4Ba-loops, i.e., β4-β5 and β8-β9loops, also play an important role in receptor binding, and hence larval toxicity [16,17]. We have also revealed that the structural stability of two receptor-binding hairpins (i.e., β2-β3 and β4-β5 within Cry4Ba-DII) through H-bonding between Thr^328^-Thr^369^ side-chains is crucial for toxin binding to the *Bt* Cyt2Aa2 toxin—an alternative receptor for Cry4Ba [18]. Moreover, we have succeeded in identifying two different types of Cry4Ba-specifc receptors from *Aedes aegypti* mosquito larvae, i.e., glycosylphosphatidylinositol (GPI)-anchored alkaline phosphatase (ALP) and GPI-anchored aminopeptidase N (APN) [19,20].

Nevertheless, the exact role of the C-terminal domain still requires more intensive investigation. Several studies have suggested that DIII could be implicated in preserving the structural integrity of Cry1Ac and Cry3Aa [7,21,22] or in determining the binding specificity of Cry1 toxins [23,24,25,26]. In addition, DIII of Cry1Ie has also been suggested to be involved in the interaction with the larval peritrophic membrane of the Asian corn borer, *Ostrinia furnacalis* [27]. In our earlier studies, the 21 kDa isolated Cry4Ba-DIII fragment was shown to be capable of binding along the apical microvilli of *A. aegypti* larval midgut, conceivably participating in toxin interactions with either lipid membranes or target protein receptors [28]. Recently, we have further demonstrated that the C-terminal domain of Cry4Ba could serve as a tight-binding anchor for lipid membrane bilayers, signifying its potential contribution to toxin–membrane interactions to mediate larval toxicity [29]. However, a precise description of toxin–receptor interactions would still need further investigation. In the present study, we have further demonstrated a pivotal role of Cry4Ba-DIII in binding to its target protein receptor—Aa-mALP. A critical residue within the Cry4Ba-DIII domain was revealed to be involved in biotoxicity against *A. aegypti* mosquito larvae, as well as in binding to the Aa-mALP receptor, further strengthening the functional involvement of Cry4Ba-DIII in toxin–receptor interactions.

## 2. Results

### 2.1. Obtaining High-Quality Purified Cry4Ba Proteins and Its Binding Partner

At the start, high-purity proteins of both the 65 kDa full-length Cry4Ba-R203Q toxin and its isolated DIII truncate were efficiently obtained (Figure 2a, lanes 1,2) through our optimized preparative procedures, which have been established earlier for each individual target protein [10,28]. It is worth mentioning that the purified Cry4Ba-DIII monomer obtained, as was previously verified by attenuated total reflection Fourier transform infrared spectroscopy [29], is more likely to adopt a distinct β-sheet structure which corresponds to its structure embodied in the full-length crystal structure, albeit detached from the N-terminal DI-DII portion (see Figure 1, inset).

Of particular interest herein, the 54 kDa Aa-mALP protein—a functional Cry4Ba receptor cloned from *A. aegypti* larval midgut [19]—was used to explore whether the Cry4Ba-DIII truncate, besides potentially contributing to toxin–lipid membrane interactions as demonstrated previously [29], is also capable of binding to a particular protein receptor. Prior to binding studies, a high-purity His_(6)_-tagged Aa-mALP protein was achieved upon urea-induced unfolding and subsequent refolding in a Ni^2+^-NTA column (Figure 2a, lane 3). Additionally, the purified refolded Aa-mALP protein was verified to preserve its phosphatase activity with an apparent specific activity of ≈0.45 µmol/min/mg toward the cleavage of the small organic phosphate pNPP, reflecting its correctly folded structure. 

### 2.2. Binding Characteristics of Cry4Ba-DIII and Its Full-Length Toxin to Aa-mALP

As revealed by a dot blot-based assay, the purified Cry4Ba-DIII truncate was able to bind to Aa-mALP immobilized on the NC membrane with detected signals apparently comparable to the full-length Cry4Ba-R203Q toxin (Figure 2b). On the contrary, no detectable signal was observed for either of the Cry4Ba proteins when the immobilized receptor was replaced with a negative control—CI-ALP, *ruling* out for non-specific binding of both protein ligands to their immobilized counterpart—Aa-mALP.

Further attempts were made to determine the binding affinity of the purified Cry4Ba-DIII truncate in comparison with that of its full-length toxin by employing ELISA-based assays. The binding data illustrated in Figure 2c revealed that the isolated DIII truncate clearly exhibited a relatively low affinity of binding to the immobilized Aa-mALP receptor with a dissociation constant (*K*_d_) of ≈1.1 × 10^−7^ M, while the smaller *K*_d_ value (≈0.8 × 10^−7^ M) was obtained for the full-length toxin. In contrast, BSA—a negative control ligand—gave a linear dose–response curve, which was indicative of no binding to the immobilized receptor.

### 2.3. Aa-mALP Homology-Based Model Supportive of Toxin-Binding Counterpart

When the conservation of amino acid residues of ALP from *A. Aegypti* and other selected organisms was analyzed through multiple sequence alignments, it displayed high similarity (>52%) among all the five different insect ALPs as being identified to be Cry toxin receptors along with the five other crystal ALP structures (see Appendix A). The pairwise alignment scores via BLAST analysis revealed the highest conservation between the Aa-mALP sequence and unsolved structure Ag-ALP sequence with 72% identity, and the highest conservation between the Aa-mALP and solved-structure shrimp-ALP sequence with 40% identity.

The resulting Ramachandran plot indicated that the 3D-modeled structure would remain in sterically favorable main-chain conformations (see Appendix A). In addition, the *z*-score of the modeled structure was −8.37, which is within the range of scores for all determined structures at a similar size and quite similar to that of the template, shrimp ALP (z-score −8.90) (Appendix A). When the modeled structure was further put to validation by VERIFY-3D, it was revealed that 93.4% of residues scored ≥ 0.2, indicating its high-quality (see Appendix A). Moreover, the C_α_-trace superposition of the Aa-mALP model and shrimp-ALP displays a 0.33 Å RMSD for about 443 equivalent C_α_ atoms out of 475 C_α_ atoms, indicating a very high structural similarity in their 3D folds (Figure 3a).

It is worth mentioning that another Aa-mALP model was also constructed based on multiple templates. When compared with the single-template model, both derived 3D models display a significant structural similarity of a C_α_-trace superposition with RMSD of 0.97 Å (see Appendix A). Therefore, such a predicted single-template model, which was built based on the best-fit shrimp-ALP template could be a suitable candidate to carry on further analysis.

As illustrated in Figure 3b, the central portion of the Aa-mALP structure (Glu^39^-Gly^513^) comprises a β-sheet core of ten strands, all but one, i.e., β15, are parallel, connected by α-helices to form a two-layer α/β sandwich, which is a typical topology of α/β hydrolase family [30]. It should be noted that the N-terminal α-helix points away from its core, and this helix was reported as a part of the dimer interface of various functional ALPs including shrimp-ALP [31]. It can also be inferred that the main divergences in these 3D molecules are found particularly in the surface-exposed residues, such as Gln^48^, Lys^155^, Thr^212^, Tyr^276^, Glu^411^, Asn^440^, and Glu^441^, which are preserved exclusively in Aa-mALP (see Figure 3c). These residues may perhaps participate in specific interactions of this receptor with its counterpart ligand Cry4Ba, although these non-conserved residues might be responsible for upholding other characteristic features.

### 2.4. Cry4Ba–Aa-mALP Docking Complex with Potential Interacting Residues

In silico docking was subsequently carried out to gain more critical insights into the architectural complex of the 65 kDa full-length Cry4Ba-R203 toxin interacting with its receptor—Aa-mALP. The best docking conformation, which was selected from more than 1000 docked models, contained the cluster of binding residues with a weighted score of -918.5 kcal/mol for center and -1035.9 kcal/mol for the lowest energy. Such a best complex structure revealed that Aa-mALP was bound to the toxin counterpart primarily through the surface-exposed loops of the receptor-binding domain—DII as well as the C-terminal domain—DIII.

MD simulations of the resulting complex were performed to refine a more precise description of dynamic interactions in such a toxin–receptor complex. The dynamic stability of the Cry4Ba–Aa-mALP complex was assessed via RMSD changes during the structural simulations, as shown in Appendix A. The plot revealed no large fluctuation in RMSD during 10ns simulations, indicating the stability of the binding complex. In addition, the RMSF (root-mean-square fluctuation) plot showed a lower RMSF value of residues corresponding to Cry4Ba-DII when comparing with that of DI and DIII, indicating the binding stability of residues in this region (see Appendix A).

The simulation results also suggested that Cry4Ba-DIII could potentially interact with Aa-mALP through at least four residues, i.e., Glu^522^ in β_16_, Asn^552^ in β_18_, Asn^576^ in a very long and unstructured loop connecting β_19_ and β_20_, and Leu^615^ in the β_22_-β_23_ loop (Figure 4, left panel inset). Additionally, Aa-mALP was bound to Tyr^322^ and Phe^364^, which are respectively located in β2-β3 and β4-β5 loops of the receptor-binding domain—DII (see Figure 4, right panel inset).

### 2.5. Biotoxicity Impairment of Cry4Ba Caused by Ala Substitution of DIII-Leu^615^

More defined experiments were further conducted to test whether the four predicted binding residues (i.e., Glu^522^, Asn^552^, Asn^576^, and Leu^615^) are potentially involved in such Cry4Ba–Aa-mALP interactions. Four Ala-substituted mutant toxins (E522A, N552A, N576A, and L615A) together with two selected Cry4Ba-DII mutant toxins (i.e., Y332A and F364A) were successfully generated using Cry4Ba-R203Q as a Wt template. When each mutant was expressed in *Escherichia coli* upon IPTG induction, all were produced as 130 kDa protoxin inclusions at levels comparable to the Wt-R203Q template toxin (Figure 5, inset). Then, experiments were performed to assess the in vitro solubility of mutant protoxin inclusions in the carbonate buffer (pH 9.0) in comparison with that of the Wt-R203Q inclusion. When the amounts of the 130-kDa soluble proteins in the supernatant were compared with those of the proteins initially used, the results revealed that all the six mutant toxin inclusions were as soluble as the Wt inclusion, giving >90% solubility. Additionally, all the mutant inclusions that can be solubilized in carbonate buffer (pH 9.0) were found to yield a 65 kDa single fragment upon tryptic digestion as similar to the Wt template (see Appendix A).

When *E. coli* cells expressing each Cry4Ba-DIII mutant were assessed for their biotoxicity against *A. aegypti* larvae, only the L615A mutant showed a large decrease in larvicidal activity (only ≈35% mortality) while the remaining mutants (i.e., E522A, N552A, and N576A) still retained their high larval toxicity (≈80–90% mortality) at levels roughly similar to Wt-R203Q (Figure 5). Nevertheless, the larval toxicity of the impaired L615A mutant was still greatly higher than that of the two Cry4Ba-DII mutant toxins, Y332A and F364A which displayed a nearly complete loss of biotoxicity. As mentioned earlier that the levels of protein expression of all the 130 kDa mutant protoxins in the form of sedimentable inclusions were roughly the same as that of the Wt-R203Q template protoxin (see Figure 5, inset). For that reason, the larvae tested in the bioassays would be considered to receive a comparable amount of the protoxin doses.

### 2.6. Exploitation of Leu^615^ in Cry4Ba-DIII for Toxin Binding to Aa-mALP

To further examine whether the functional importance of DIII-Leu^615^ for larval toxicity is involved in such receptor–toxin interactions, the L615A bio-impaired mutant toxin along with the two bio-inactive DII-mutant toxins (Y332A and F364A) were then tested for their binding capability to Aa-mALP in comparison with the Wt-R203Q template toxin. Before being studied by ELISA-based binding analysis, a 65 kDa high-purity protein of all the selected mutant toxins were productively obtained (see Figure 6a, inset). It should be noted that in our study, we would simply focus on functional binding and thus the three other DIII-mutant toxins (i.e., E522A, N552A, and N576A) that did not show a significant decrease in larvicidal activity were excluded from further binding analysis.

As can be seen in Figure 6a, the relative binding results revealed that L615A exhibited >60% reduction in binding activity for the immobilized target receptor as compared to the Wt-R203Q template toxin set at 100% relative binding. Nonetheless, both Y332A and F364A mutants, which almost completely lost their biotoxicity, were found to retain an apparently higher degree of binding activity against Aa-mALP (≈60–70% relative binding) while BSA, a negative control ligand, showed only marginal or no binding to the immobilized receptor.

It should be also noted that in silico attempts have been made via the web-based BeAtMuSiC program [32] in order to calculate the binding affinity change (ΔΔG_bind_) in Cry4Ba–Aa-mALP interactions upon Ala substitutions. Such calculation revealed that the L615A mutation exhibited the largest change of binding free energy (1.54 kcal/mol) as compared to five other mutations (E522A, N552A, N576A, Y332, and F364) giving ΔΔG_bind_ of only 0.20–0.56 kcal/mol (see Figure 6b).

## 3. Discussion

According to our previous studies via immunohistochemical staining, the 21 kDa cloned Cry4Ba-DIII fragment, which was over-expressed in *E. coli* as a soluble monomeric form, was apparently able to bind to the apical microvilli of *A. aegypti* mosquito-larval midgut cells [28]. Recently, such a Cry4Ba-DIII truncate was shown to be tightly bound to immobilized liposome membranes, exhibiting the dissociation rate constant (*k*_off_) comparable to the 65 kDa full-length toxin [29]. In the present study, further efforts were made via more critical approaches in the interest of a precise mechanistic role of the Cry4Ba-DIII truncate in binding to its target receptor protein—Aa-mALP—and hence potential contribution to larval toxicity.

As initially revealed by dot blotting, the isolated Cry4Ba-DIII fragment, besides serving as a tight-binding anchor for lipid membrane bilayers, could also play a role in specific binding of the Cry4Ba toxin to the NC-immobilized Aa-mALP receptor. This perception is consistent with many other studies for lepidopteran-active Cry toxins as suggesting that DIII could play a significant role in receptor binding, particularly in the specificity determination of different insect species [25,26,33,34]. For instance, DIII of either Cry1Ab or Cry1Ea has been demonstrated by replacement with Cry1Ca-DIII to generate hybrid Cry1Ab-Cry1Ca or Cry1Ea-Cry1Ca toxins with altered characteristics of target insect specificity [26]. Cry1Ab-DIII has also been shown to be involved in functional interactions with ALPs or APNs in different susceptible insect larvae [33,34]. Nonetheless, further determination of binding affinity via sandwich ELISA revealed that Cry4Ba-DIII exhibited a rather weak binding to the immobilized Aa-mALP protein with an apparent *K*_d_ value of ≈1.1 × 10^−7^ M as compared with the full-length toxin (*K*_d_≈0.8 × 10^−7^ M). Such a weaker binding affinity (i.e., higher *K*_d_ value) of the isolated truncate correlates well with our previous observation viaimmuno-histochemical assays that the Cry4Ba-DIII fragment displayed a lower binding signal than the full-length toxin toward *A. aegypti* larval midgut apical microvilli [28]. In addition, a modest binding activity of the DIII truncate would indicate a functional requirement of DII—the receptor-binding domain for the most efficient cooperative binding to Aa-mALP.

It has been noted that DIII of the Cry1A toxins shares high overall structural similarity with the carbohydrate-binding domain of several proteins, e.g., 1,4-β-glucanaseCenC, β-glucoronidase, galactose oxidase, and sialidase (for reviews, see [35]). Additionally, DIII of the Cry1A toxins was shown to bind specifically to an *N*-acetyl galactosamine (GalNAc) moiety present on APN receptors of lepidopteran insect larvae [36,37]. Moreover, GalNAc on ALP receptors was demonstrated to be essential for the binding of the lepidopteran-specific Cry1Ac toxin to its target ALPs from *Heliothis virescens* and *Helicoverpa armigera* larvae [38,39]. However, these previous findings [36,37,38,39] seem inconsistent with our present data, which suggested that a saccharide moiety is not exclusively required for the binding of either Cry4Ba full-length toxin or its DIII truncate to the Aa-mALP receptor because the Aa-mALP protein expressed in *E. coli* cells is most unexpected to be glycosylated, even if it has now become evident that bacteria can acquire both protein *N*- and *O*-glycosylation pathways [40,41,42]. Our notion is in good agreement with other previous studies which also suggested that an *E. coli* expressed-*An. gambiae* ALP isoform (Ag-ALP1t) was not glycosylated and thus, that its counterpart toxin—Cry11Ba—was supposed to recognize the polypeptide part rather than a sugar portion [43].

Thus far, there is no experimentally resolved 3D structure of the Aa-mALP receptor or other insect ALPs; therefore, a plausible homology-based model of Aa-mALP was built based on the highest sequence similarity of the known shrimp-ALP structure. The Phi/Psi values observed in the modeled structure signify that such an obtained Aa-mALP model would remain in sterically favorable main-chain conformations. Through successive docking between the Cry4Ba toxin and its counterpart—Aa-mALP—and followed by structural simulations, we have identified a best docking complex structure that could infer at least four Cry4Ba-DIII residues (i.e., Glu^522^, Asn^552^, Asn^576^, and Leu^615^) to be potentially implicated in such toxin–receptor interactions. Subsequent Ala substitutions of the four individual residues clearly disclosed that only the L615A mutant exhibited a severe reduction in toxicity against *A. aegypti* larvae. Thus, these results suggested that DIII-Leu^615^ located in the β_22_-β_23_ loop (see Figure 7) is basically involved in Cry4Ba activity against the target mosquito larvae, conceivably being exploited for the toxin binding to Aa-mALP.Nevertheless, it is noteworthy that not all the receptor-interacting residues predicted by such in silico global docking showed an impaired effect on toxin activity when they were individually mutated to Ala.

The functional importance of DIII-Leu^615^ was strengthened by further binding analysis via ELISA, which revealed that the L615A-impaired toxin clearly displayed a decrease in binding capability to the immobilized Aa-mALP receptor. It was unexpected that the two bio-inactive DII-mutant toxins, Y332A and F364A, which almost totally lost their larval toxicity, apparently retained a higher degree of binding activity than the DIII-L615A mutant toxin. This may suggest a certain extent of non-functional binding of Cry4Ba-DII to Aa-ALP via either Tyr^332^ or Phe^364^. Another possible explanation for Y332A and F364A mutations that showed a modest effect on toxin binding to the Aa-mALP receptor is that both Cry4Ba-DII residues would be exploited for the binding to other target receptors on *A. aegypti* larval cell membrane rather than to Aa-mALP. In fact, the DII of numerous Cry toxins have been widely demonstrated to play a pivotal role in specific binding to a variety of target insect receptors, e.g., GPI-ALPs, GPI-APNs, cadherin-like proteins (CLPs), and hence larval toxicity [8,9]. Therefore, the Cry4Ba-DII domain could possibly bind to GPI-APN, CLP, or perhaps others as-yet-unidentified receptors.

Consistent with such ELISA-based binding results, the calculated binding affinity changes (ΔΔG_bind_) in Cry4Ba–Aa-mALP interactions of the two DII residue mutations were also found to be substantially lower than that of the L615A mutation, which showed the largest ΔΔG_bind_ among all the Ala substitutions. Thus, these ΔΔG_bind_ data would also support a functional importance of DIII-Leu^615^. Accordingly, besides the receptor-binding domain—DII, the C-terminal domain—DIII of Cry4Ba could also serve as a potential receptor-binding moiety in which such a bulky hydrophobic residue, DIII-Leu^615^, is likely to be exploited for specifically interacting with a hydrophobic core made by Pro^42^ and Trp^45^ on its target insect receptor—Aa-mALP (see Figure 7, left panel). However, there was a diminished contactor a cavity occurred inside such an interacting cluster of the L615A mutant protein (see Figure 7, right panel), and hence a disruption of toxin binding and biotoxicity. It was previously noted that Leu^511^ of Cry1Ab-DIII was critically involved in functional binding to ALP—a target receptor from *Manduca sexta* larvae [44].

As was also noted in a variety of proteins that the Leu residue could act as a versatile binding moiety potentially involved in protein–protein interactions, depending on its location and side-chain orientation. For example, Leu^31^ of phospholamban, a homopentameric transmembrane protein in the sarcoplasmic reticulum (SR), was suggested to be essential for productive interaction with the Ca^2^^+^pump of cardiac SR [45]. In addition, Leu^135^ of tropomodulin-1, a well-defined actin-capping protein was shown to play a critical role in binding to the erythrocyte tropomyosin [46]. Another study demonstrated that the single Leu^20^ residue within the activation domain of the oncogenic protein E2A-PBX1 is required in the effective interaction with a hydrophobic pocket within the CREB-binding domain of the cyclic AMP response element binding (CREB) protein [47].

## 4. Materials and Methods 

### 4.1. Preparation of the Cry4Ba Active Toxin and Its DIII Truncate

The 65 kDa Cry4Ba active toxin was prepared from *E. coli* strain JM109 expressing the 130 kDa Cry4Ba-R203Q protoxin in which one trypsin-cleavage site at Arg^203^ was mutated to Gln, thus giving a 65 kDa activated toxin upon tryptic digestion, as described previously [48]. Toxin preparation was accomplished by proteolytic digestion of the protoxin pre-solubilized in carbonate buffer (50 mM Na_2_CO_3_/NaHCO_3_, pH 9.0) with trypsin (*N*-tosyl-L-phenylalanine chloromethyl ketone-treated, Sigma-Aldrich, Burlington, VT, USA) before being subjected to purification by size-exclusion FPLC (fast protein liquid chromatography using Superose^®^12, HR10/30, GE Healthcare Bio-Sciences, Piscataway, NJ, USA) as described previously [10].

For preparation of the 21 kDa DIII fragment, the cloned Cry4Ba–DIII truncate, which was over-expressed as a soluble form in *E. coli* strain JM109 under control of the *lac* promoter, was effectively purified by anion-exchange (Resource Q column, GE Healthcare Bio-Sciences, Piscataway, NJ, USA) and size-exclusion FPLC as described elsewhere [28]. Both purified proteins were analyzed by sodium dodecyl sulfate-(12% *w*/*v*) polyacrylamide gel electrophoresis (SDS-PAGE) prior to the quantification of protein concentrations using the Bradford microassay (Bio-Rad, Hercules, CA, USA).

### 4.2. Construction and Preparation of Cry4B Mutant Toxins

The p4Ba-R203Q plasmid, encoding the 130 kDa Cry4Ba–R203Q protoxin [48], was used as a template together with individual pairs of complementary mutagenic primers (see Appendix A) designed for single Ala substitutions of selected DIII residues. All mutant plasmids were generated by polymerase chain reaction using high-fidelity Phusion DNA polymerase (Finnzymes, Espoo, Southern Finland, Finland), following the QuickChange^TM^ mutagenesis procedure (Stratagene, Santa Clara, CA, USA). Mutant plasmids in selected *E. coli* clones were initially identified by restriction endonuclease analysis and subsequently verified by DNA sequencing. All mutant toxins were over-expressed in *E. coli* strain JM109 and purified as described earlier [10].

### 4.3. Expression and Purification of Aa-mALP—Cry4Ba Toxin Receptor

Upon IPTG induction, the cloned Aa-mALP protein (lacking the secretion signal and GPI attachment sequences) fused with 6×His-tag at its C-terminus was over-expressedas an inclusion in *E. coli* strain BL21(DE3) under control of the *T_7_* promoter. A 54 kDa purified His_(6)_-tag fused Aa-mALP protein was efficiently obtained viaurea-induced denaturation (8M) and subsequent renaturation in a Ni^2+^-nitrilotriacetic acid (NTA) affinity column (HisTrap HP column; GE Healthcare, Chicago, IL, USA) by gradients of decreasing urea concentrations as described elsewhere [19].

### 4.4. ALP Activity Assay of the Purified His-Tagged Aa-mALP Protein

The purified His_(6)_-tag fused Aa-mALP protein was assayed for its phosphatase activity using *p*-nitrophenyl phosphate (pNPP) (New England BioLabs, Ipswich, MA, USA) as a colorimetric substrate. The hydrolysis of 5 mM pNPP to *p*-nitrophenol (pNP), a water-soluble yellow product with strong absorbance at 405 nm in ALP buffer (5 mM MgCl_2_, 100 mM NaCl, and 100 mM Tris-HCl, pH 9.5) was assessed at 25 °C, using a microplate spectrophotometer. The specific activity of ALP was expressed as the release of pNP in µmole/min/mg of protein.

### 4.5. Dot Blot-Based Binding Assay of Immobilized Aa-mALP with Its Ligands

The refolded Aa-mALP (2.5 µg) and calf intestinal ALP (CI-ALP—a negative control, New England BioLabs, Ipswich, MA, USA) were directly dotted onto a nitrocellulose (NC) membrane. The dotted membrane was blocked with phosphate-buffered saline (PBS, pH 7.4) containing 5% skim milk for 1 h and subsequently incubated with the activated Cry4Ba-R203Q toxin or its DIII truncate (200 nM) in PBS–5% skim milk for 30 min. After the membrane was washed with PBS, pH 7.4/Triton X-100 (3 times, a total of 15 min), it was incubated with rabbit anti-Cry4Ba antibodies (1:10,000 dilution) in PBS–5% skim milk for 1 h. The immuno complexes were subsequently detected with horseradish peroxidase (HRP)-conjugated anti-rabbit IgGs (1:5000 dilution, Cell Signaling Technology, Danvers, MA, USA) followed by color development with 3,3’,5,5’-tetramethylbenzidine (TMB, Vector Lab, Burlingame, CA, USA) and H_2_O_2_.

### 4.6. Receptor-Binding Affinity Assay of Cry4Ba Toxins 

Quantitative measurements of binding affinity of the purified Cry4Ba-DIII fragment to an immobilized counterpart receptor—Aa-mALP were performed in comparison with that of its 65 kDa full-length toxin, using sandwich enzyme-linked immunosorbent assay (ELISA)-based method as described previously [18] with some modifications. Then, 96-well Maxi-binding microplates (SPL Life Science, Pocheon, Gyeonggi, South Korea) were coated with purified Aa-mALP (2.5 µg) in 50 mM carbonate buffer (pH 9.0) at 4 °C for 4 h. After blocking with 5% skim milk in PBS, pH 7.4/Triton X-100, the coated wells were incubated with each individual purified Cry4Ba protein or a negative control ligand—bovine serum albumin (BSA) at different concentrations (100, 200, 300, and 400 nM) at 37 °C for 2 h. The ligand proteins bound to the immobilized Aa-mALP were detected by probing sequentially with rabbit anti-Cry4Ba antibodies (1:10,000 dilution), followed with HRP-conjugated anti-rabbit IgGs (1:5000 dilution). Color was developed with TMB substrate/H_2_O_2_, yielding a yellow-colored product with an absorbance at 450 nm recorded using an automated microplate reader (Multiskan, MTX Lab systems, Vienna, VA, USA). Plots of absorbance versus toxin concentration were constructed for determining the dissociation constant (*K*_d_) of toxin interactions with immobilized Aa-mALP. Statistical analysis via Student’s *t*-test was carried out for determining significant differences of the relative binding of all the tested mutants compared with that of Wt toxin binding to its immobilized receptor.

### 4.7. Biotoxicity Assays of Cry4Ba and Its Mutant Toxins 

Bioassays for the larvicidal activity of *E. coli* cells expressing Cry4Ba wildtype (Wt) or its mutant toxins were performed against *A. aegypti* mosquito larvae as described elsewhere [49]. Cells containing pUC12 vector plasmid were used as a negative control. Mortality was recorded after 24 h incubation period, and three independent experiments were performed for each toxin. Statistical analysis was carried out by using Student’s *t*-test to determine significant differences between the larvicidal activities of mutants and that of Wt.

### 4.8. Protein Multiple Sequence Alignments of Insect ALPs 

The deduced amino acid sequence of Aa-mALP [50] was aligned with four homologous insect ALPs (*An. gambiae*, Ag-ALP (EAA10738); *Heliothis virescens*, Hv-ALP (ABR88230); *Bombyx mori*, Bm-ALP (BAA14420) and *Culex quinquefasciatus*, Cq-ALP (XP_001868288)) together with that of ALP structures from other organisms (Human placenta, human-pALP (PDB: 1EW2); *E. coli*, *E. coli*-ALP (PDB: 1ED9); Antarctic bacterium, TAB5-ALP (PDB: 2W5V); *Vibrio* sp., *Vibrio*-ALP (PDB: 3E2D); Archaea bacterium *Halobacterium salinarum*, Archaea-ALP (PDB: 2W0Y) and shrimp, shrimp-ALP (PDB: 1K7H)), using ClustalW through Clustal server (www.ebi.ac.uk/Tools/msa/clustalw2/), followed by manual refinement.

### 4.9. Homology-Based Modeling and Validation of Aa-mALP Structure

A plausible three-dimensional (3D) Aa-mALP model was generated based on the target-template alignment with the shrimp-ALP high-resolution [1.9Å] crystal structure [PDB: 1K7H] [31] via SWISS-MODEL homology modeling (http://swissmodel.expasy.org). The pairwise sequence alignment between Aa-mALP and shrimp-ALP was analyzed via compositional score matrix adjustment of theNCBI’s protein-query protein-database BLAST (Basic Local Alignment Search Tool) program (https://blast.ncbi.nlm.nih.gov/Blast.cgi?PAGE=Proteins).A multi-template Aa-mALP model was generated using the Modeller 10.1 [51] based on crystal structures of shrimp-ALP (PDB: 1K7H), human-pALP (PDB: 1EW2), and Antarctic bacterium, TAB5-ALP (PDB: 2W5V).

The local quality of the single-template modeled structure was validated by QMEAN 6 [52]. VERIFY-3D [53] was used to assess the compatibility of an atomic model with its own amino acid sequence, while ProSA [54] was applied to test the energy criteria of the modeled Aa-mALP in comparison with experimentally PDB structures. The error of the homology model structure was analyzed by ERRAT [55], and its stereochemistry was analyzed using the Psi/Phi Ramachandran plot computed with PROCHECK [56]. Finally, the 3D-modeled structure was refined by molecular dynamics (MD) simulations as described in more details below. The root mean square deviation (RMSD) of the Aa-mALPmodel to the shrimp–ALP template structure was calculated by structural superposition using ChimeraTool [57].

### 4.10. In Silico Intermolecular Docking between Cry4Ba-R203Q and Aa-mALP

Molecular docking was done using ClusPro 2.0 [58] in order to identify the binding sites between the Aa-mALP model and the Cry4Ba activated form (PDB: 1W99) as their coordinate files were generated by SWISS-MODEL (http://swissmodel.expasy.org). Docking calculations were performed using ClusPro 2.0 through a global soft rigid body search program (PIPER). The top 30 largest clusters were obtained from the 1000 best energy conformations, and the best docking complex structure, containing the largest cluster of residues, was subjected to energy minimization and then MD simulations to get more precise in binding structure. Bonding between interacting residues was analyzed as hydrogen bonding was calculated based on the criteria of 2.7 Å for hydrogen–accepter distance and 3.3 Å for donor–acceptor distance using LIGPLOT 1.4.4 [59]. Electrostatic interactions were computed with criteria of distance cut-off 4.0 Å between oxygen and nitrogen using an extension of the molecular visualization VMD program, version 1.8.7 [60]. Hydrophobic interactions were calculated using LIGPLOT 1.4.4 based on default criteria of 2.9 Å and 3.9 Å for minimum and maximum contact distance, respectively. A possible effect of Ala substitutions on toxin–receptor interactions of each mutant–receptor docking complex was analyzed by computing the binding affinity changes (ΔΔG_bind_) using the BeAtMuSiC program [32].

### 4.11. Structural Simulations of Aa-mALP and Cry4Ba–Aa-mALP Docking Complex

The protein structure file (PSF) for Aa-mALP was generated using an automatic PSF builder extension of the VMD software [60] with CHARMM27 force field. The Aa-mALP model was solvated in 116 × 92 × 105 Å^3^ water box with a 15 Å buffering distance. Then, 28 sodium (Na^+^) and 17 chloride (Cl^—^) ions were placed randomly into the system for charge neutralization and achieving a concentration of 150 mM NaCl. The final system of the modeled structure has 80,144 atoms consisting of 7112 atoms of Aa-mALP, 24,309 water molecules, 28 Na^+^ions, and 17 Cl^—^ ions. For the Cry4Ba–Aa-mALP docking complex structure, the complex was solvated in a 116 × 146 × 112 Å^3^ water box. Then, 37 Na^+^ and 27 Cl^—^ions were added randomly. The final system contains 152,675 atoms consisting of 15,977 atoms of the Cry4Ba–Aa-mALP complex, 45,524 water molecules, 37 Na^+^ions, and 27 Cl^—^ions.

All MD simulations were conducted using NAMD 2.9 [61], CHARMM27 force field, as well as the TIP3P model for liquid water [62]. At first, an energy minimization process was performed for 1 ps and then heated to 300 K for 5 ps, which was equilibrated at pressure 1 atm with all heavy atoms of protein under harmonic constraints using a force constant of 2.5 kcal/mol/Å^2^. Then, equilibration without constrains was performed for 10 ns. The simulations were performed using periodic boundary conditions. Temperature and pressure were controlled via Langevin dynamics and Nose–Hoover Langevin piston, respectively.

## Figures and Tables

**Figure 1 toxins-13-00553-f001:**
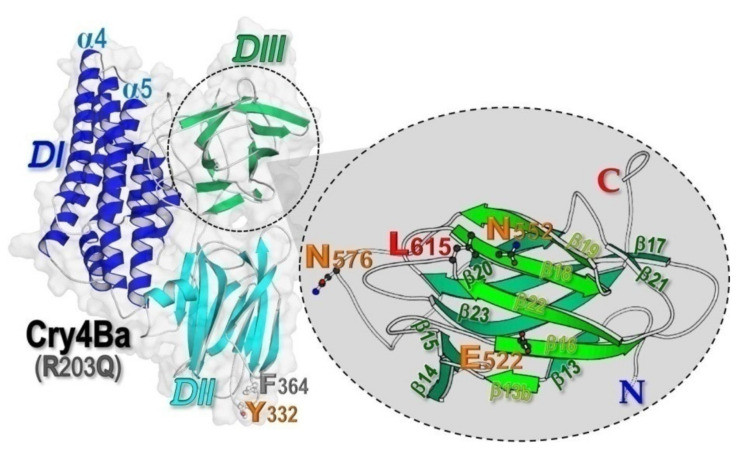
Combined surface-ribbon representation of the 65 kDa Cry4Ba-R203Q crystal structure (PDB: 4MOA [10]) prepared via MolScript, illustrating its three-domain organization (DI–DIII). Inset, ribbon representation of a different view of the C-terminal β-sheet sandwich DIII, illustrating a predicted Aa-mALP-interacting residues (Glu^522^, Asn^552^, Asn^576^, and Leu^615^).

**Figure 2 toxins-13-00553-f002:**
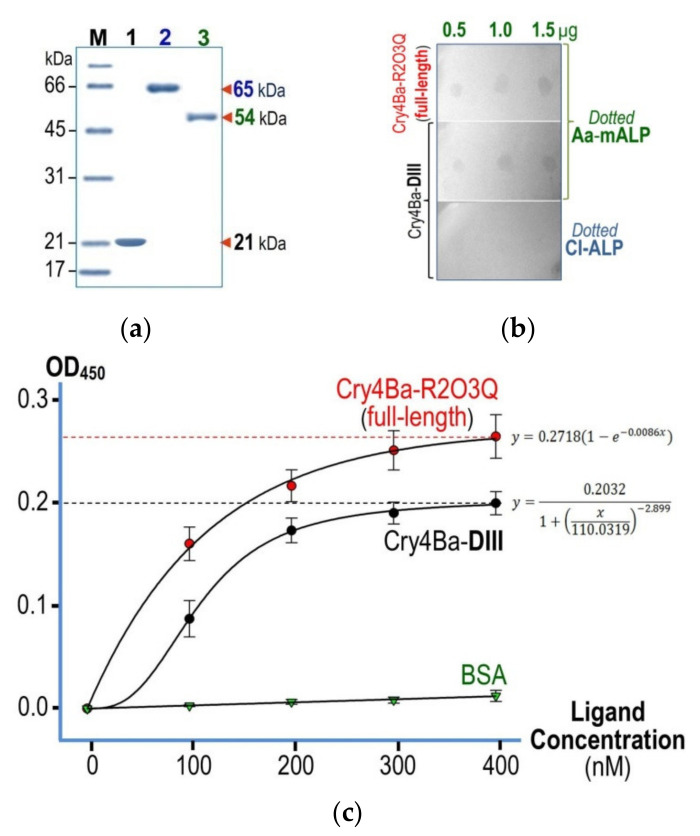
(**a**) SDS-PAGE (Coomassie brilliant blue-stained 12% gel) analysis of purified target proteins, the 21 kDa Cry4Ba-DIII protein (lane 1), the 65 kDa Cry4Ba-R203Q full-length toxin after trypsin activation and FPLC purification (lane 2), and the 54 kDa His tag-fused Aa-mALP protein purified by elution through a Ni^2+^-NTA column. M, molecular mass standards. (**b**) Dot blot-based binding analysis of the Cry4Ba-R203Q full-length toxin and the isolated DIII protein against the NC-immobilized Aa-mALP protein (0.5, 1.0, and 1.5 µg). Immobilized CI-ALP (0.5, 1.0, and 1.5 µg) was used as a negative control. (**c**) Dose–response curves for Aa-mALP receptor binding of the full-length toxin and its DIII truncate protein analyzed by sandwich-ELISA. BSA was used as a negative control ligand. Error bars indicate standard errors of the mean (SEM) from three independent experiments.

**Figure 3 toxins-13-00553-f003:**
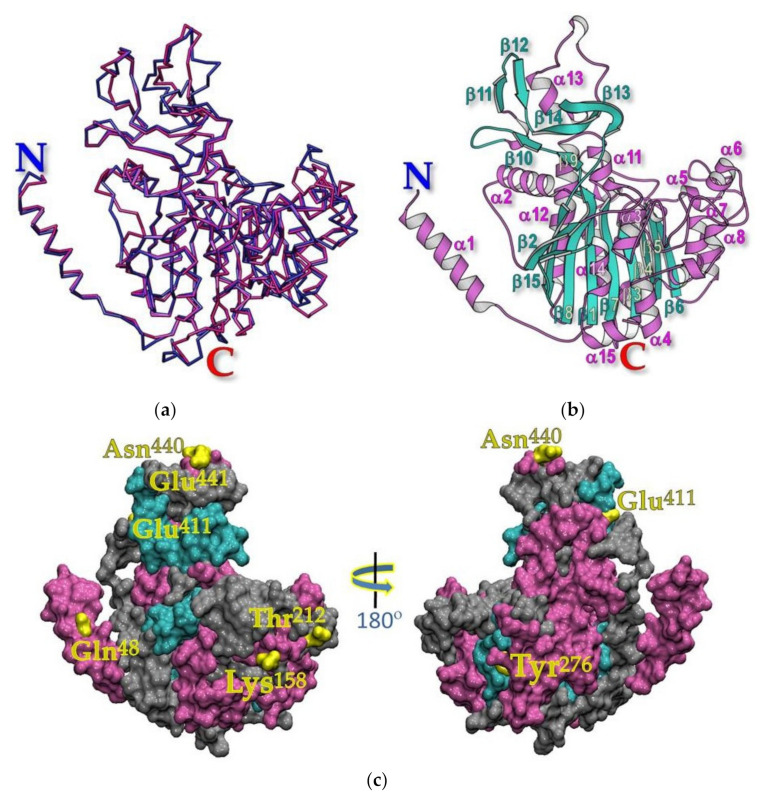
Homology-based 3D modeled structure of Aa-mALP. (**a**) Superposition of Cα traces of the Aa-mALP model (blue) with shrimp ALP (deep pink) structures prepared via Chimera 1.7.(**b**) Ribbon representation of Aa-mALP modeled structure prepared via MolScript. Ten β-strands (cyan) surrounded by α-helices (pink) in the core structure. (**c**) Surface representation of the modeled Aa-mALP structure prepared via VMD software. Surface-exposed residues unique to Aa-mALP are shown in yellow while those in α-helices, β-strands, and connecting loops are shown in pink, cyan, and gray, respectively.

**Figure 4 toxins-13-00553-f004:**
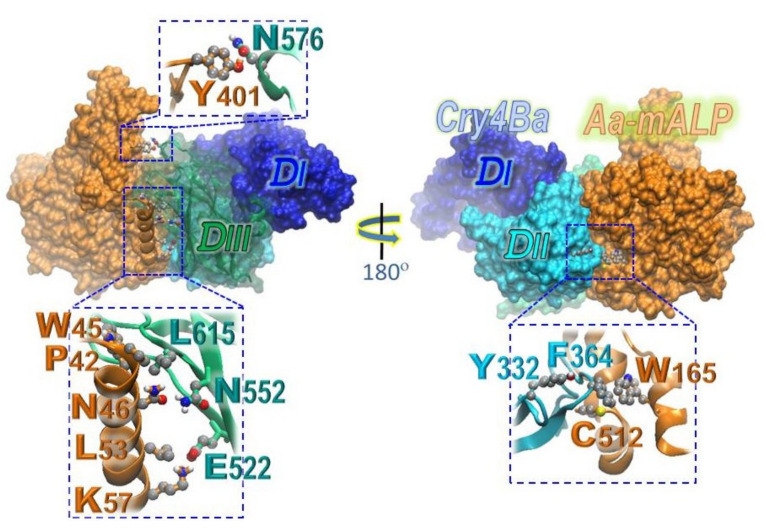
Surface representation of the resulting MD-simulated docking complex prepared via Chimera 1.7, illustrating the most promising conformation of the three-domain Cry4Ba-R203Q toxin interacting with its receptor—Aa-mALP. The potential receptor-binding residues in Cry4Ba-DIII (i.e., Glu^522^, Asn^552^, Asn^576^, and Leu^615^) as well as those in DII (i.e., Tyr^332^ and Phe^364^) are represented as ball-and-stick models along with their interacting partner residues.

**Figure 5 toxins-13-00553-f005:**
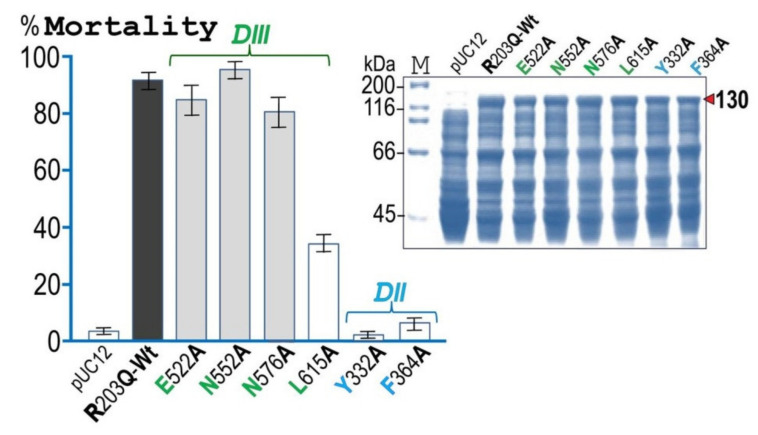
Larvicidal activity of *E. coli* cells (≈10^8^ cells/mL) expressing the Cry4Ba-R203Q or its DIII (i.e., E522A, N552A, N576A, and L615A) and DII (i.e., Y332A and F364A) mutants tested against *A. aegyptilarvae*. Cells containing the pUC12 plasmid vector were used as a negative control. Error bars indicate SEM from at least three independent experiments. Inset, SDS-PAGE (Coomassie brilliant blue-stained 12% gel) analysis of the lysate extracts of *E. coli* cells (≈10^7^ cells)expressing the 130 kDa Cry4Ba-R203Q or its mutant protoxins.

**Figure 6 toxins-13-00553-f006:**
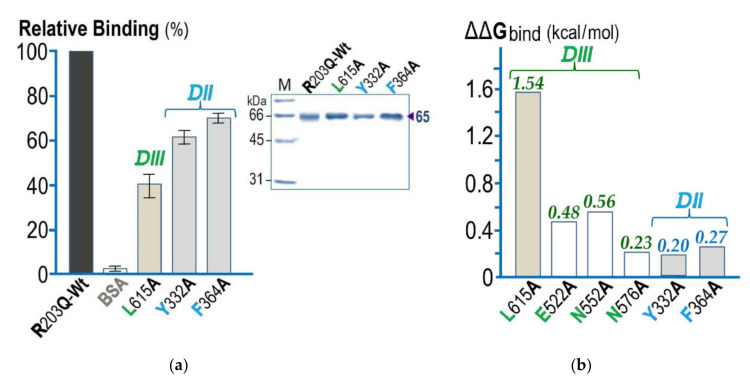
(**a**) Analysis of relative binding activity against Aa-mALP of three selected mutants (DIII: L615A, DII: Y332A, and F364A), which display a decrease in larval toxicity. Binding activity of the Cry4Ba-R203Q-Wt template toxin was taken as 100% and percentage of relative binding for each tested mutant toxin. BSA was used asa negative control ligand. Error bars indicate SEM from three independent experiments. Inset, SDS-PAGE (Coomassie brilliant blue-stained 12% gel) analysis of the 65 kDa R203Q-Wt and its three mutant toxins after trypsin activation and FPLC purification as indicated. M, molecular mass standards. (**b**) In silico calculation of the change of binding free energy (ΔΔG_bind_) in Cry4Ba–Aa-mALP interactions upon each mutation as indicated using the BeAtMuSiC program [32].

**Figure 7 toxins-13-00553-f007:**
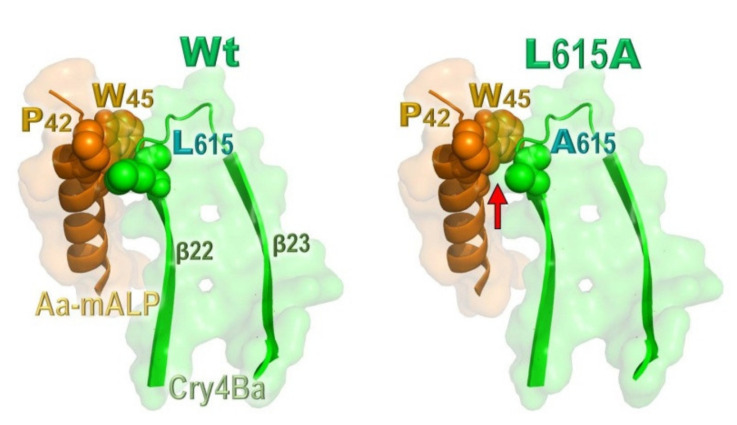
Combined surface-ribbon representation of part of the Cry4Ba–Aa-mALP docking complex, illustrating the critical interaction between DIII-Leu^615^ and a potential hydrophobic core made by Pro^42^ and Trp^45^ on Aa-mALP (left panel) along with a diminished contact (denoted by an arrow) in such a toxin–receptor interaction occurring in the Cry4Ba-L615A mutant (right panel).

## Data Availability

Not applicable.

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
