# Peer review of "Bacillus thuringiensis Cry4Ba Insecticidal ToxinExploits Leu615 in Its C-Terminal Domain to Interact with a Target Receptor—Aedes aegypti Membrane-Bound Alkaline Phosphatase"

_toxins, 2021, doi:10.3390/toxins13080553_

Round 1

Reviewer 1 Report

The authors of this study focus on the interaction between Cry4Ba Insecticidal Toxin and its putative target Aa-mALP to investigate the mechanism of action between the two proteins. The initial experimental data is convincing that the constructs used for the experiments are valid and that the DIII domain is likely one of the domains that interact with the target protein. However, the computational analyses used for the study are not rigorous enough to explain the following experimental results that demonstrated the role of L615 in binding and bioactivity of the toxin. I believe that the subject of the research is interesting and the results would contribute to the knowledge in the toxins field, but a significant rework of the computational section is necessary. Below are my point-by-point comments for the different aspects of computational modeling applied in the paper. I hope the authors find these comments helpful.

Homology modeling

  • The real metric of whether homology models converged to a single native-like model is score versus RMSD plots. The authors should include this data to support the claim that their model is a realistic representation of the target protein. 1000 models may or may not be sufficient for their purposes, the only way to know is to see convergence.
  • Homology modeling can benefit from the use of multiple templates, therefore all available structural information should be used at the homology modeling step instead of just using a single model. Therefore, the authors should build a model using all the different templates and compare this model to that of reported in this study. If the RMSD values between the two models and the conformations of the significant residues are similar, then the current model is also acceptable, given that they show convergence in plots described in Point 1.

Docking

  • For global docking what kind of filtering/calculations were done to decide on a particular binding site? Global docking is notoriously vague and should be used with caution. Was any experimental data used to validate these sites?
  • If the interaction surface is similar to what is proposed by the authors, then all the mutations should have an effect in theory. The authors should explain why such an effect is not observed in further detail and based on their calculations. I do not find the current level of explanation satisfactory. Please see point 3 for a likely solution.
  • The activity of the peptides and their binding are two discrete events based on the experimental results. Docking can only answer the former question as the activation/inactivation event can be more complex and may involve the movement of structural domains of the protein. In order to draw conclusions on the effect of alanine mutations, their effects should be tested first in silico. The authors need to quantify the effect of mutations in terms of docking score changes associated with the residue to alanine mutations. While there are multiple ways of doing this, the most precise and practical methods would be running ΔΔG calculations through molecular dynamics simulations or a more advanced docking software. The disadvantage of the latter is that the deviations can be so large that the energy changes may not be captured. Whatever is the method of selection, there should be a new figure that shows the peptides interacting with all the proposed point mutations along with the calculated binding energies.

Molecular dynamics

  • The simulation times are too short to draw any meaningful conclusions regarding RMSD or RMSF, but they could be okay for pure side chain refinement. At least three calculations of 200 ns should be run for each system investigated to be able to draw meaningful conclusions from the RMSD and RMSF calculations.
  • The RMSF values look chaotic because they are not baselined or normalized. If the authors choose to reproduce this data with longer calculations, Figure 4A right panel should be normalized by dividing by average RMSF or something in that vein to make the values of the two proteins more comparable.
  • If the mutation calculations will be run through MD simulations, each should be run to get an acceptable average of the complex system. Again, the calculations should be longer to the order described in point 1 for such calculations.

Minor things

  • A high-resolution zoomed in figure clearly showing the environments of L615, F364, and Y332 at the WT complex are necessary. These can be merged into a single figure.
  • Figure 4A and 4B are irrelevant and should be two separate figures. If the RMSD and RMSF values are not put in context as now, they can be moved to the supporting section.
  • P2 l52, the PDB ID of the structure should be noted and cited.

Author Response

Please see an attachment.

Reviewer 2 Report

The paper entitled “Bacillus thuringiensis Cry4Ba Insecticidal Toxin Exploits Leu615 in Its C-terminal Domain to Interact with a Target Receptor—Aedes aegypti Membrane-Bound Alkaline Phosphatase” presents information on the Cry4Ba mosquito-specific toxine especially on its domain III. Several biochemical analysis (binding assays, mutagenesis and biotoxicity) were conducted and demonstrated its role in receptor binding and consequently in toxic activity. This molecular approach based on the modification of some crucial residue provides significant results to understand the mode of action on this toxin family which turns out to be complex. Abstract and Introduction are well written. The results are clear and well discussed.

Most of my remarks relate to the material and methods used for this study.

  1. I don't understand why the larvicidal activity was performed with the coli lysates since the different recombinant proteins can be obtained and purified quite easily (figure 6). Also, the authors affirmed that comparable amount of toxin doses were used. I m not convinced since more R203Q wt protein can be observed compared to the load of F364A toxin.

  1. No statistical analysis has been performed to compare the relative binding of all mutants. The binding activity of DIII mutant is significant different compared to that of DII mutants.

Addressing these points may require minor revisions, but may improve the quality of the manuscript

Author Response

Please see an attachment.

Round 2

Reviewer 1 Report

Please see the attached file with my comments. 

Author Response

Please see an attachment.

Round 3

Reviewer 1 Report

With the DDG calculations to support the authors' conclusions and with the acknowledgements regarding the shortcomings of the study added to the manuscript, I believe it is publishable in its current state in terms of scientific content.